# Uncovering the Multifaceted Role of *PA2649* (*nuoN*) in Type III Secretion System and Other Virulence Production in *Pseudomonas aeruginosa* PAO1

**DOI:** 10.3390/microorganisms13020392

**Published:** 2025-02-11

**Authors:** Lin Chen, Yujie Si, Xue Han, Yue Xiao, Yidan Pan, Kangmin Duan, Songzhe Fu

**Affiliations:** 1Key Laboratory of Resource Biology and Biotechnology in Western China, Ministry of Education, College of Life Science, Northwest University, Xi’an 710069, China; siyujie@stumail.nwu.edu.cn (Y.S.); hanxue@stumail.nwu.edu.cn (X.H.); xiaoyue@stumail.nwu.edu.cn (Y.X.); panyidan@stumail.nwu.edu.cn (Y.P.); 2Department of Medical Microbiology and Infectious Disease, Rady Faculty of Health Sciences, University of Manitoba, Winnipeg, MB R3E 0W2, Canada; kduan@nwu.edu.cn; 3School of Medicine, Northwest University, Xi’an 710069, China

**Keywords:** *Pseudomonas aeruginosa*, type III secretion system, NADH dehydrogenases, motility, antibiotic resistance

## Abstract

*Pseudomonas aeruginosa* is a multi-drug-resistant opportunistic pathogen that adapts to challenging environments by deploying virulence factors, including the type III secretion system (T3SS). Emerging evidence points to a role for NADH dehydrogenase complexes in regulating virulence; however, their precise contributions remain unclear. Here, we identify *PA2649*, a component of the NADH dehydrogenase complex I (*nuo* operon), as a key regulator of T3SS-related activities. *PA2649* deletion resulted in a twofold increase in *exoS* expression and enhanced cytotoxicity in both A549 cell and Chinese cabbage models. Full revertant of the *nuo* operon was necessary to restore *exoS* expression to wild-type levels, suggesting a critical connection between NADH dehydrogenase activity and T3SS regulation. The *PA2649* mutation also disrupted the Rsm-Exs regulatory axis, downregulating *gacS*, *rsmY*, *rsmZ*, and *hfq* while upregulating *exsC*. Overexpression of *rsmY*, *rsmZ*, *gacA*, *hfq*, and *exsD* partially rescued T3SS function, confirming that *PA2649* influences T3SS via the Rsm-Exs pathway. Furthermore, *PA2649* deletion altered motility, biofilm formation, pyocyanin production, protease activity, and antibiotic susceptibility. These phenotypes could not be complemented with T3SS regulatory genes alone, indicating that *PA2649* modulates these traits through mechanisms independent of the Rsm-Exs axis, potentially involving NADH dehydrogenase-associated pathways. This study underscores the multifaceted role of *PA2649* in regulating *P. aeruginosa* pathogenicity and resistance, providing novel insights into its complex regulatory networks and highlighting new avenues for therapeutic targeting.

## 1. Introduction

Bacteria can sense a wide range of physical and chemical external cues, adapting their metabolic processes accordingly to survive in diverse environments. Pathogenic bacteria, in particular, detect both physical (e.g., surface properties) and chemical signals (e.g., Ca^2+^, N-acyl homoserine lactones (AHL), 2-heptyl-3-hydroxy-4-quinolone (*Pseudomonas* Quinolone Signal, PQS), cyclic di-GMP etc.) during infection, enabling them to modulate virulence factors either directly, through mechanisms such as two-component systems, or indirectly, via metabolic adjustments that enhance pathogenicity [1]. While numerous regulatory proteins, including quorum sensing regulators, two-component system regulators, and sigma factors, have been identified as critical in controlling virulence factors at both transcriptional and post-transcriptional levels [2,3], the mechanisms by which metabolic genes influence virulence factor expression remain incompletely understood.

T3SS, first identified in *Yersinia* in the 1990s [4], is a virulence-associated protein complex that facilitates the direct injection of bacterial proteins into host cells, thereby enhancing pathogenicity through trans-kingdom communication with eukaryotic hosts [5]. T3SS is commonly found in various in many pathogenic bacteria, including *Yersinia*, *Salmonella*, *Shigella*, *Bordetella*, *Pseudomonas*, and enteropathogenic *Escherichia* species [6]. The regulation of T3SS continues to be a significant area of research due to its essential role in bacterial virulence and intercellular interactions.

*Pseudomonas aeruginosa*, a versatile pathogen, is capable of causing infections in various tissues, including the respiratory tract, urinary system, and bloodstream, particularly in immunocompromised individuals [7]. The T3SS of *P. aeruginosa*, which consists of several structurally and functionally conserved proteins, can severely compromise host defenses by injecting effector toxins such as ExoU, ExoT, ExoS, and ExoY [8,9]. These toxins disrupt phagocytosis, facilitate immune evasion, and promote colonization. T3SS serves as a model for studying both transcriptional and post-transcriptional mechanisms that regulate T3SS gene expression in bacterial pathogens.

Given its ability to infect a diverse range of hosts (mammals, insects, worms, amoebae, and plants), *P. aeruginosa* relies on multifaceted regulatory systems for precise spatial and temporal control of its T3SS. The expression of T3SS is finely tuned by various environmental factors such as extracellular calcium, host proteins, abiotic surfaces, nucleoid-associated proteins, and two-component and global regulatory systems [10,11,12]. The core regulatory pathway, which employs a partner-switching mechanism, includes the key genes *exsA*, *exsC*, *exsD*, and *exsE*, which are involved in the transcription activation of the aforementioned classical effectors (ExoS, ExoT, ExoU, and ExoY) [7]. Upon induction, ExsE is secreted extracellularly, enabling ExsC to bind to ExsD and release ExsA [13]. The liberated ExsA then directly binds to target promoters to activate the T3SS regulon [14]. Additional layers of regulation involve the Rsm system, cAMP, cyclic di-GMP signaling, and other global regulatory systems (Such as quorum sensing (QS) systems and the two-component regulatory systems like GacS-GacA) [12,15]. The Rsm system, including small RNAs (RsmY and RsmZ) and regulatory proteins (RsmA/RsmF), modulates T3SS and other cellular processes, integrating environmental and metabolic signals to optimize the expression of virulence factors [16].

The metabolic state significantly influences the T3SS, as studies have shown that a proficient Proton Motive Force (PMF) is essential for effector secretion [17]. Metabolic defects, such as those observed in the *crc* mutant, lead to a reduction in T3SS activity [18]. Additionally, NADH metabolism also appears to be linked to virulence in *P. aeruginosa*, with NADH/NAD⁺ and NADPH/NADP⁺ playing crucial roles in redox reactions that are vital for cellular metabolism and biosynthesis [19]. These studies offer valuable insights into the role of NADH-related genes in the regulation of virulence. *Pseudomonas aeruginosa* expresses three NADH dehydrogenases (NUO, NQR, and NDH-2), which oxidize cellular NADH and serve as entry points for electrons in the respiratory chain [20,21,22]. This adaptability, combined with the central role of NADH in bacterial physiology, underscores the necessity of investigating the role of NADH dehydrogenases in the pathogenicity of *P. aeruginosa*, as this connection remains largely unexplored.

Here, our findings demonstrate that the deletion of *PA2649* in *P. aeruginosa* significantly upregulates *exoS* expression and cytotoxicity while disrupting multiple virulence traits, including motility, biofilm formation, and pyocyanin production. The *PA2649* gene influences T3SS regulation via the Rsm-Exs pathway, yet its impact on antibiotic resistance and other virulence phenotypes appears to involve separate mechanisms beyond this pathway.

## 2. Materials and Methods

### 2.1. Bacterial Strains and Culture Conditions

Bacterial strains and plasmids used in this study are presented in Appendix A. Primers used in this study are showed in Appendix A. *Pseudomonas aeruginosa* and *Escherichia coli* were cultured on Luria–Bertani (LB) agar or in LB broth at 37 °C unless otherwise specified. Antibiotics were used at the following concentrations: for *E. coli*, kanamycin (Kan) at 50 µg/mL, ampicillin (Amp) at 100 µg/mL, tetracycline (Tc) at 15 µg/mL, and gentamicin (Gen) at 15 µg/mL in LB. For *P. aeruginosa*, Gen at 50 µg/mL in LB or 150 µg/mL in *Pseudomonas* isolation agar (PIA; Beijing Land Bridge Tech., Ltd., Beijing, China), Tc at 70 µg/mL in LB or 300 µg/mL in PIA, carbenicillin (Cb) at 250 µg/mL in LB, and trimethoprim (Tmp) at 300 µg/mL in LB. Other antibiotics, such as chloramphenicol (Chl), ciprofloxacin (Cip), streptomycin (Strep), and polymyxin B (PMB), were used in LB as indicated. Peptone and Yeast extract were purchased from OXOID (Hampshire, UK). Antibiotics used in this study were all purchased from MP Biomedicals (Shanghai, China).

### 2.2. Construction and Revertant of Gene Knockout Mutants

Following previously established methods, we employed homologous recombination using the sucrose-lethal *sacB* gene for knockout mutant construction [23]. To delete the *PA2649* gene, we amplified 1575 bp and 976 bp fragments from the regions upstream and downstream of the *PA2649* open reading frame, respectively, via PCR. These PCR products were ligated into the pEX18Tc plasmid, which was subsequently transferred into the *E. coli* conjugation donor strain S17-1 *λpir* and then introduced into wild-type PAO1 by biparental mating [24]. Single-crossover mutants, with the plasmid integrated into the PAO1 chromosome, were selected using tetracycline. These mutants were cultured in PIA medium (300 µg/mL Tc) overnight and then plated on 15% sucrose plates to select for double-crossover mutants. Mutants were verified by PCR across the target gene region. For constructing revertant and overexpression strains, the promoter and ORF of target genes were cloned into the digested pAK1900 vector, which was then transformed into the corresponding strains.

### 2.3. Construction of T3SS Gene Expression Reporters

The plasmid pMS402, which carries a promoterless *luxCDABE* reporter gene cluster, was used to construct promoter-*luxCDABE* gene fusions as previously described [25]. Promoter regions of target genes were amplified by PCR using high-fidelity Pfu DNA polymerase and primers designed based on the PAO1 genome sequence. The amplified promoter regions were then cloned into the *Bam* HI–*Xho* I sites upstream of the *lux* genes on pMS402 [26]. The resulting plasmids were introduced into PAO1 by electroporation, and cloned promoter sequences were confirmed by DNA sequencing. In addition to the plasmid-based reporter system, a chromosomal reporter system was constructed using the integration plasmid CTX6.1, derived from the mini-CTX-*lux* plasmid [26]. CTX6.1 contains all elements required for chromosomal integration, replication origin, and tetracycline resistance. The *luxCDABE* reporter cassette from pMS402, including the kanamycin resistance marker and multiple cloning sites, was isolated and ligated into CTX6.1. The generated plasmid was first transferred into *E. coli* SM10-*λ pir*, and integration into PAO1 was achieved by biparental mating [27]. Gene expression using these *lux*-based reporters was quantified in liquid culture as counts per second (cps) of light production, measured with a Synergy H1 microplate reader (BioTek, Winooski, VT, USA) [28].

### 2.4. Gene Expression Detection

Gene expression in liquid cultures was measured using *lux*-based reporters, quantified as cps of light production with a Synergy H1 microplate reader (BioTek, Winooski, VT, USA) [29]. Overnight bacterial cultures were inoculated at 1% into fresh LB medium containing appropriate antibiotics and incubated for 3 h. For measurement, 95 µL of LB and 5 µL of bacterial culture were added to each well of a black 96-well plate with a clear bottom, mixed thoroughly, and topped with 50 µL of paraffin oil to prevent evaporation. Luminescence (cps) and bacterial growth (OD_600_) were recorded every 30 min over a 24 h period using a multimode microplate reader, as previously described.

### 2.5. Cytotoxicity Assay

Cytotoxicity assays were conducted with slight modifications as previously described [30]. Bacterial cytotoxicity was assessed by measuring the survival of A549 cells following infection with *P. aeruginosa*. A549 cells were cultured in DMEM supplemented with 10% (*v*/*v*) fetal bovine serum, Amp (100 mg/mL), and Strep (100 mg/mL) at 37 °C with 5% CO_2_. Approximately 5000 cells were seeded per well in a 96-well plate and cultured overnight until reaching 80–90% confluence. The culture supernatants were removed, and the cell monolayers were washed once with phosphate-buffered saline (PBS). Overnight bacterial cultures were diluted in fresh LB medium and grown at 37 °C for approximately 12 h to an OD_600_ of 1.0. The cultures were centrifuged at 12,000× *g* for 2 min, and the supernatant was filtered through a 0.22 µm membrane. A549 cells were incubated with 25 µL of bacterial supernatant and 75 µL of fresh DMEM per well. After a 3 h infection period, 10 µL of Cell Counting Kit-8 (CCK-8) solution (Biosharp, Hefei, China) was added to each well and incubated for an additional hour. OD_450_ was measured using a Synergy H1 microplate reader (BioTek, Winooski, VT, USA) to calculate cell survival. The cell survival rate (%) was calculated as follows: Cell survival rate (%) = [(OD_450_ of treated sample − OD_450_ of blank)/(OD_450_ of untreated control − OD_450_ of blank)] × 100.

### 2.6. Extraction of ExoS Proteins by Trichloroacetic Acid (TCA) Precipitation

ExoS proteins were extracted using TCA precipitation as previously described [31]. *Pseudomonas aeruginosa* cultures were grown under T3SS-inducing conditions (5 mM EGTA, 20 mM MgCl_2_) for 6 h at 37 °C. After removing bacterial cells by centrifugation at 12,000× *g*, 110 µL of 100% TCA was added to the supernatant and placed on ice for 10 min. Subsequently, 500 µL of 10% TCA was added, and the mixture was kept on ice for an additional 20 min to precipitate proteins. Following protein precipitation, the supernatant was removed by centrifugation at 20,000× *g* for 30 min. The protein pellet was washed with acetone and then pelleted again. Proteins were resuspended in buffer, separated by SDS-PAGE, and visualized with Coomassie blue staining.

### 2.7. Chinese Cabbage Infection Assay

The cabbage infection assay was performed as previously reported with minor modifications [32]. Bacterial strains were cultured overnight at 37 °C and harvested by centrifugation. The bacterial pellets were washed with 10 mM MgSO_4_, the supernatant was removed, and the cells were resuspended. The OD_600_ of the bacterial suspension was adjusted to 2.0. Cabbage stems, pretreated with 0.1% H_2_O_2_, were placed in Petri dishes containing filter paper soaked in 10 mM MgSO_4_. A 10 µL aliquot of the bacterial suspension was injected into each cabbage stem, and the samples were incubated at 30 °C for 6 days. The extent of cabbage tissue decay was then observed and recorded.

### 2.8. Biofilm Assay

The biofilm formation assay was conducted as previously described with minor modifications [33]. Overnight cultures of *P. aeruginosa* were diluted to an OD_600_ of 0.1 in fresh LB medium. Aliquots of 200 µL of the diluted culture were added to the wells of a 96-well polystyrene microtiter plate. The plates were incubated at 37 °C for 24 h to allow biofilm formation. After incubation, the culture medium was carefully aspirated, and the wells were washed three times with sterile PBS to remove nonadherent cells. The biofilm was fixed by adding 200 µL of 100% methanol to each well and incubating at room temperature for 15 min. The methanol was then aspirated, and the wells were air dried. To visualize the biofilm, 200 µL of 1% crystal violet solution was added to each well and incubated for 20 min. After staining, the plates were washed gently with PBS to remove excess dye, and the biofilm was solubilized by adding 200 µL of 95% ethanol to each well. The absorbance of each well was measured at 570 nm using a microplate reader.

### 2.9. Pyocyanin Assay

Pyocyanin production was measured as previously described [34]. Overnight cultures were inoculated at 1% into 5 mL of PB medium and incubated at 37 °C with shaking at 200 g for 16 h. After incubation, cells were removed by centrifugation, and the supernatant was subjected to organic extraction with 3 mL of chloroform. Following centrifugation, the lower chloroform layer was transferred to a new tube, where 1 mL of 0.2 M HCl was added, mixed thoroughly, and allowed to separate into layers. The absorbance of the upper aqueous phase was measured at 520 nm to quantify the pyocyanin concentration.

### 2.10. Swimming, Swarming, and Twitching Motility Assays

Bacterial motility was assessed as previously described [35]. The swimming medium comprised 0.5% NaCl, 1% tryptone, and 0.3% agar. The swarming medium contained 0.8% nutrient broth, 0.5% glucose, and 0.5% agar. For twitching, the medium consisted of 0.5% yeast extract, 1% tryptone, 1% NaCl, and 1.2% agar. *Pseudomonas aeruginosa* cultures were diluted to an OD_600_ of 0.5, and 2 µL of the diluted culture was placed centrally on the surface of each agar plate. Plates were incubated at 37 °C for 24 h, after which the diameter of the motility zone was measured. For twitching motility, the bacterial suspension was stabbed into the bottom of the medium and incubated at 37 °C for 24 h. The diameter of the twitching zone was assessed using crystal violet staining.

### 2.11. Protease Assay

Protease activity was assessed using a milk medium containing 2.5% skimmed milk powder and 1% agar to detect extracellular protease activity, as previously described [36]. Overnight bacterial cultures were diluted to an OD_600_ of 0.2, and 2 µL of each bacterial suspension was spotted onto the surface of the milk medium. Plates were incubated at 37 °C for 12 h. Protease activity was determined by comparing the diameter of the clear zones surrounding each bacterial spot, indicating proteolytic degradation of the milk proteins.

### 2.12. Antibiotic Resistance Determination

Antibiotic susceptibility assays were performed as previously described with minor modifications [36]. Overnight bacterial cultures were inoculated into fresh LB medium at 1% and incubated for 3 h. LB media with various antibiotic concentrations were prepared, and 95 µL of each antibiotic-containing medium was combined with 5 µL of bacterial culture in each well of a 96-well plate. To prevent evaporation, 50 µL of liquid paraffin was added to each well. Bacterial growth was monitored every 30 min for 24 h by measuring OD_600_ with a multimode microplate reader. The minimum inhibitory concentration (MIC) was defined as the lowest antibiotic concentration at which no bacterial growth was detected after 24 h. Additionally, antibiotic resistance assays were performed on LB agar plates with various antibiotic concentrations. Overnight cultures were serially diluted, and 5 µL of each dilution was spotted onto the corresponding antibiotic-containing plates. The plates were incubated at 37 °C for 24 h, and bacterial growth was then observed.

### 2.13. Extraction and Quantification of Intracellular NADH and NAD^+^

NADH and NAD^+^ were extracted using the Beyotime Enhanced NAD^+^/NADH Assay Kit (WST-8) (Beyotime, Shanghai, China) following the manufacturer’s instructions (Cat No: S0176S). Bacterial strains were cultured overnight in fresh LB medium for approximately 12 h. The OD_600_ of the bacterial cultures was adjusted to the same level before processing. A 300 µL aliquot of each bacterial culture was centrifuged at 12,000× *g* for 5 min, and the supernatant was discarded. Subsequently, 200 µL of the NAD^+^/NADH extraction buffer was added, and the pellet was gently resuspended to facilitate cell lysis. The mixture was then centrifuged at 12,000× *g* for 10 min at 4 °C. The resulting supernatant was collected and used as the sample for analysis.

### 2.14. Statistical Analysis

Data were analyzed and visualized using GraphPad Prism, version 9.5. All experiments were conducted in triplicate and independently repeated three times. Statistical significance was assessed using two-tailed unpaired *t*-tests where applicable, with significance thresholds set as follows: ns (not significant, *p* > 0.05); * *p* < 0.05; ** *p* < 0.01; and *** *p* < 0.001.

## 3. Results

### 3.1. Upregulation of exoS Expression and Enhancement of Pathogenicity in ∆nuoN

During the preliminary screening for mutants related to the expression of the type III secretion system, several transposon mutants were identified that could potentially affect its expression. We selected one transposon insertion mutant within the *PA2649* gene (Appendix A) for further investigation. First, we constructed a *PA2649* deletion mutant and integrated an *exoS* luminescent reporter into the mutant’s genome using a CTX-based system [26]. Expression analysis (Figure 1A) showed that *exoS* expression in the ∆*PA2649* mutant was more than two times higher than in the wild-type strain. Next, to verify that the observed phenotype was indeed caused by the *PA2649* mutation, we constructed a *PA2649* revertant plasmid with a fragment A (Figure 1B) in pAK1900 and introduced it into the mutant. However, the results showed that *exoS* expression did not revert to wild-type levels (Figure 1C). According to the literature [37], the NADH oxidase activity lost due to a *nuoN* gene mutation could be restored by a plasmid containing the previously identified *nuoN* gene along with the upstream intergenic region between *nuoM* and *nuoN* in *Escherichia coli.* Therefore, we constructed four additional revertant plasmids with different genes (named B–E), as shown (Figure 1B), and introduced them into the mutant to assess *exoS* expression. The results showed that only the plasmid complementing the entire operon (pAK1900 with E fragment in Figure 1B) was able to restore the *exoS* expression (Figure 1A). None of the other plasmids could restore the phenotype of the *PA2649* mutant (Figure 1C), suggesting that the mutation in *PA2649* led to a loss of function in the entire *nuo* operon, resulting in the upregulation of *exoS* expression. Additionally, in order to eliminate the polar effect on *PA2650*, we constructed the pAK-*PA2650* plasmid and introduced it separately into both the *PA2649* mutant and PAO1 to examine *exoS* expression. Our results indicated that the overexpression of *PA2650* in the *PA2649* mutant had no effect on *exoS* expression (Appendix A), whereas the overexpression of *PA2650* in PAO1 led to approximately a onefold reduction in *exoS* expression (Appendix A).

In summary, the results above indicate that the *PA2649* mutation leads to elevated *exoS* expression. To further verify this finding, we extracted proteins from the corresponding culture supernatants and compared the results, which showed that the protein levels in the mutant were higher than those in the wild-type strain (Figure 2A). Due to the increased expression of the *exoS* caused by the *PA2649* mutant, we utilized A549 cells as an in vitro model to assess the cytotoxicity of the mutant. The results of cytotoxicity assays using the supernatant from the *PA2649* mutant showed a significant increase in cytotoxicity compared to the wild type and complemented strains (Figure 2B). Furthermore, the mutant exhibited a slightly more pronounced phenotype with enhanced tissue damage in a Chinese cabbage model (Figure 2C).

### 3.2. PA2649 Influences the Type III Secretion System Through the Rsm-Exs Regulatory Axis

To further explore whether *PA2649* exerts its effects through known regulatory pathways, we investigated its potential involvement in the Rsm-Exs regulatory axis, a major pathway controlling T3SS. Literature suggests that this axis plays a crucial role in regulating T3SS expression [12]. Consequently, we analyzed the expression levels of several genes within the Rsm-Exs axis. Our results indicated a reduction in the expression of *gacS*, *rsmY*, *rsmZ*, *hfq*, and an increased expression of *exsC* (Figure 3). In contrast, the expression levels of *rsmA*, *retS*, *exsD*, *PA1611*, and *gacA* remained unchanged (see Figure 3).

ExsD coding antiactivator which ExsC preferentially binds resulting in the release of ExsA, liberated ExsA then binds to target promoters and recruits RNA polymerase to activate the T3SS regulon [38]. To validate these findings, we first overexpressed the *exsD* gene in the *PA2649* mutant, which successfully restored the mutant phenotype (Figure 4A). Similarly, the overexpression of the upstream regulators *rsmY* and *rsmZ* partially restored the mutant phenotype, albeit to varying extents (Figure 4B,C). Given that GacA is known to activate *rsmYZ* expression, we hypothesized that overexpressing *gacA* in the *PA2649* mutant would restore the T3SS phenotype. As predicted, our results confirmed this hypothesis (Figure 4E). In previous research, we discovered that PA1611 interacts with RetS to regulate T3SS expression via the Rsm-Exs axis [30]. To further confirm this interaction, we overexpressed *PA1611* in the *PA2649* mutant, which successfully complemented the mutant phenotype (Figure 4D). However, the overexpression of other genes, such as *ladS*, failed to restore the mutant phenotype (Figure 4F). GacSA is a two-component regulatory system, with GacS acting as a membrane-bound sensor kinase that phosphorylates GacA, thereby activating GacA to regulate the expression of other genes [39]. Unexpectedly, the overexpression of *gacS* in the *PA2649* mutant not only failed to restore the mutant phenotype but also resulted in a slight increase in *exoS* expression compared to the wild type (Figure 4H).

These results presented above suggest that *PA2649* modulates T3SS expression via the Rsm-Exs pathway. Accordingly, we hypothesized that the deletion of *rsmA* in the *PA2649* mutant background would substantially suppress T3SS expression. In alignment with this hypothesis, T3SS expression was markedly reduced in the ∆*PA2649* ∆*rsmA* double mutant, exhibiting a phenotype comparable to that of the *rsmA* single mutant and markedly lower T3SS expression than observed in the wild type (Figure 5).

We further validated the above results using cytotoxicity assays. Consistent with the expression data, the cytotoxicity of the supernatants from the overexpression of *ladS* and *gacS* in *PA2649* mutant, which failed to complement *exoS* expression, was similar to that of the *PA2649* mutant. In contrast, the cytotoxicity of strains that successfully complemented *exoS* expression was restored to wild-type levels (Figure 6).

### 3.3. Phenotypic Effects of the PA2649 Mutation on Other Virulence Factors

The above results demonstrate that the expression of genes such as *gacS* and *rsmYZ* is significantly altered in the *PA2649* mutant. These genes are key regulatory components within the Rsm pathway, which ultimately exerts its effects through RsmA. To further investigate the impact of these changes, we assessed additional related phenotypes. While swimming motility showed no significant difference compared to the wild-type strain (Figure 7A), both twitching and swarming motilities were markedly reduced (Figure 7B,C). Biofilm adhesion was significantly decreased at 6 and 12 h (Figure 7D), though no changes were observed at 24 h (Figure 7D). Pyocyanin and proteinase production was also significantly reduced (Figure 7E,F).

Since the expression of *exoS* in the *PA2649* mutant is regulated by the Rsm-Exs axis, we naturally considered whether the observed phenotypes are also regulated by this axis. The results showed that the overexpression of *PA1611*, *hfq*, *gacA*, *rsmY*, *rsmZ*, *ladS*, and *gacS* in the *PA2649* mutant did not restore the protease, pyocyanin production, or motility-related phenotypes (Figure 8). This suggests that the impact of the *PA2649* mutation on protease, pyocyanin, and motility is regulated via a pathway distinct from the one controlling *exoS* expression.

### 3.4. Role of PA2649 in Perturbation of Central Metabolism and Antibiotic Resistance

NADH dehydrogenases are commonly found in energy-transducing membranes, where they serve as key sites for initiating electron transport. As an electron acceptor, NAD^+^ is critical for energy metabolism and cellular processes since it directly impacts an organism’s capacity to generate energy and maintain metabolic homeostasis. The dynamic interconversion between NAD^+^ and NADH maintains cellular redox balance and influences various physiological processes. Therefore, we tested whether the ratio of NAD^+^ and NADH was changed in the mutant by using the commercial NAD^+^/NADH assay kit. A single disruption of the *nuoN* gene resulted in an increase in the intracellular NAD(H/+) pool (Figure 9A) and the ratio of NAD^+^ and NADH compared to the control strain (Figure 9B). The changes in the NAD^+^/NADH ratios and total concentrations of NAD(H/+) were primarily due to changes in NAD^+^ production, as the NADH concentrations were relatively constant.

In addition, previous research has shown that the MIC of gentamicin for the ∆*nuoIJ* and ∆*nuoG* mutants was increased compared to the wild-type strain PA14 [20,21]. Therefore, we investigated whether the ∆*nuoN* mutation alters sensitivity to various antibiotics. Our results indicated a significant reduction in sensitivity to Chl, with the MIC of PAO1 at 120 μg/mL, and the MIC for the *nuoN* mutant increased threefold to 480 μg/mL (Table 1). Additionally, we observed a onefold increase in resistance to Gen, with the MIC rising from 1.5 mg/mL in the wild-type to 3 mg/mL in the mutant (Table 1), which is consistent with previously reported trends [20]. Moreover, the *nuoN* mutant exhibited approximately a onefold increase in resistance to Cip (Table 1), with MICs increasing from 0.3125 μg/mL to 0.625 μg/mL for Cip (Table 1). No changes in resistance were observed for Kan, Strep, and PMB (Table 1). The complemented strain restored sensitivity to wild-type levels (Table 1 and Appendix A). Similarly to protease activity, pyocyanin production, or motility-related phenotypes, revertants of ∆*PA2649* with *PA1611*, *hfq*, *gacA*, *rsmY*, *rsmZ*, *ladS*, and *gacS* failed to restore the mutant’s susceptibility (Appendix A). This suggests that the effect of the *PA2649* mutation on antibiotic resistance is also regulated through a pathway distinct from that controlling *exoS* expression.

## 4. Discussion

In the present study, we demonstrated that in *P. aeruginosa*, the *PA2649* (*nuoN*) mutation may affect *exoS* expression via the Rsm-Exs axis and influence other phenotypic traits through distinct pathways (Figure 10).

To investigate the effect of the *nuoN* mutation on T3SS regulation, we constructed a *nuoN* deletion mutant and observed a significant upregulation of *exoS*. Restoration of wild-type *exoS* levels required revertant with the entire *nuo* operon, whereas partial or gene-specific complementation was ineffective (Figure 1C), indicating that the *nuoN* mutation likely disrupts the full function of the NUO complex. Previous studies found minimal impacts on NADH dehydrogenase activity in certain NUO mutants [20]; thus, we did not directly measure this activity here. Instead, our results confirm that *exoS* overexpression specifically results from the *PA2649* mutation. Further experiments showed that the overexpression of the full *nuo* operon in PAO1 did not suppress *exoS* expression (Appendix A), supporting the hypothesis that *exoS* upregulation in the mutant is due to the loss of *nuoN* rather than the overall activity of the *nuo* operon. Cytotoxicity assays with mutant supernatants revealed increased cell toxicity (Figure 2B), consistent with elevated *exoS* levels (Figure 1A). Reduced pyocyanin in the mutant supernatant suggests that other virulence factors, likely proteinaceous, contribute to the observed cytotoxicity, as evidenced by decreased toxicity upon protease K treatment (Appendix A). Torres et al. reported that *P. aeruginosa* NADH dehydrogenase deletion mutants (Δ*nuoIJ* and Δ*ndh*) displayed slightly reduced tissue damage and fewer recoverable CFUs in a lettuce model [20]. Moreover, in an insect model (*Galleria mellonella* waxworm model), the kinetics of killing was significantly slower in the ∆*nuoIJ* strain compared to PAO1, though the LD50 was unchanged with the loss of NDH-1 [20]. The strain lacking NUO (Δ*nuoG*) showed similar macrophage-killing ability as the wild type, while two other NADH dehydrogenase mutants (Δ*nqrF* and Δ*ndh*) demonstrated significantly greater cytotoxicity [21]. These findings seem to contradict our results, but differences in experimental models and mutated genes may account for this discrepancy. This phenomenon is also evident in our observations on antibiotic resistance (see below), suggesting that mutations in different subunits of the NADH dehydrogenase complex can produce distinct phenotypic outcomes.

In *P. aeruginosa*, T3SS regulation is predominantly controlled by ExsA, an AraC/XylS family transcription factor, whose expression is regulated by the P*_exsC_* and P*_exsA_* promoters, with P*_exsC_* being more active under inducing conditions [14,40,41]. In the *nuoN* mutant, we observed increased P*_exsC_* promoter activity without changes in *exsD* levels, supporting a model wherein T3SS regulation occurs via the Exs system. Given the upstream complexity of Exs regulation [30,42,43], we examined the expression levels of related regulatory factors, including *rsmA*, *rsmZ*, *rsmY*, *gacA*, *gacS*, *retS*, *ladS*, and *PA1611*, all of which influence T3SS expression via the Rsm system [39,44,45]. We detected significant reductions in *rsmZ*, *rsmY*, and *gacS* expression in the mutant, while *retS*, *ladS*, *PA1611*, *gacA*, and *rsmA* remained stable (Figure 3). Phenotype restoration through RsmYZ, PA1611, or GacA revertant further underscores the role of the RsmYZA pathway in *PA2649*-mediated T3SS regulation. Interestingly, revertant with Hfq also restored the mutant phenotype (Figure 4G), suggesting potential regulation via either the Rsm system or direct modulation of ExsA. Hfq, as reported, can stabilize RsmY to indirectly modulate ExsA and Vfr levels [46]. Additionally, Hfq interacts with multiple T3SS-related mRNAs, though its precise regulatory role remains to be fully elucidated [47]. Its involvement here implies a multi-faceted regulatory network that warrants further investigation.

Our data show that GacA successfully complemented the ∆*PA2649* phenotype, whereas *gacS* and *ladS* overexpression did not, highlighting regulatory distinctions between these components that are also documented in related studies [48,49]. The differential effects of overexpressing GacS and GacA are likely due to their distinct roles and regulatory mechanisms within the two-component system. The overexpression of GacS alone may increase the input signal sensitivity but could saturate the system without sufficient GacA to propagate the signal. On the other hand, the phosphorylation signal from GacS is not appropriately modulated since the regulation of GacS in *P. aeruginosa* is intricately modulated by multiple interacting proteins, reflecting the complexity of its role in environmental adaptability and virulence. Overexpressing *gacS* in a plant-beneficial bacterium (*Pseudomonas chlororaphis* HT66-FLUO) restored the biosynthesis of PCN, whereas *gacA* overexpression did not, suggesting that the regulatory targets of GacA and GacS differ [50]. In the *PA2649* mutant strain, GacS expression is significantly reduced (Figure 3). This lower expression level likely results in diminished GacS signaling capacity, potentially disrupting the finely tuned balance between activation and inhibition mediated by LadS and RetS. The observed reduction in GacS expression in the *PA2649* mutant suggests that *PA2649* plays a previously uncharacterized role in maintaining optimal GacS levels. This reduced expression may exacerbate the effects of regulatory interactions, tipping the balance of activation and inhibition in favor of lower GacS activity. Further investigations into the *PA2649*-GacS interaction, along with a detailed analysis of how LadS, RetS, and RsmA contribute to GacS signaling under different environmental conditions, will provide deeper insights into this complex regulatory network. Furthermore, our results suggest that GacA functions downstream of *PA2649* in the T3SS pathway, as evidenced by GacA revertant restoring T3SS function in a Δ*PA2649* background, while *PA2649* revertant did not in a Δ*gacA* background (Appendix A). These findings support a model in which *PA2649* primarily regulates T3SS via the Gac-Rsm-Exs axis, although additional pathways, such as Vfr-cAMP, Fis, FleQ, or Crc, may also contribute to this regulatory network.

Antibiotic susceptibility testing of the *nuoN* mutant revealed increased resistance to Chl, Cip, and Gen, but no significant changes in resistance to PMB, Kan, or Strep. This antibiotic profile suggests alternative resistance mechanisms, likely including MexEF-OprN, which mediates resistance to both chloramphenicol and ciprofloxacin [51]. Full chloramphenicol resistance requires effective uptake and additional mechanisms [32]. Future studies should determine whether chloramphenicol resistance in the *nuoN* mutant is linked to reduced permeability or another similar efflux pump, such as the ABC extrusion system (PP2669/PP2668/PP2667) or the AgmR regulator (PP2665) in *P. putida* KT2440 [52]. By contrast, the *nuoG* mutant, as reported by Rreha et al. [21], displayed increased kanamycin resistance and reduced chloramphenicol sensitivity, underscoring mutation-specific regulatory effects and suggesting that different subunits of NADH dehydrogenase may distinctly influence resistance phenotypes. NUO and NDH-2 function as the primary NADH dehydrogenases during aerobic growth in LB, with polymyxin B known to inhibit NDH-2 activity [53], which may explain the lack of change in polymyxin B resistance in the *nuoN* mutant. Additionally, the overexpression of genes involved in T3SS regulation, including PA1611, GacA, RsmY, RsmZ, and ExsD, did not restore motility, pyocyanin production, or antibiotic resistance in the ∆*nuoN* mutant, suggesting that the phenotypic changes in this mutant arise from pathways distinct from those regulated by the T3SS system. The NuoN protein, as part of the NADH dehydrogenase I complex, has been proposed to implicate in the maintenance of PMF [20], which is critical for efflux pump activity in *P. aeruginosa*. Efflux pumps, such as MexAB-OprM, are known to contribute to resistance against β-lactam antibiotics [54], including penicillin-based therapies. Although our study does not directly investigate this relationship, the potential role of *nuoN* in modulating efflux pump function and antibiotic resistance deserves further exploration. This perspective underscores the multifaceted impact of *nuoN* on both virulence and resistance phenotypes, highlighting its importance as a potential target for therapeutic intervention.

Functionally, NADH dehydrogenase, particularly *nuoN* (a subunit of Complex I), serves as a primary electron entry point in the respiratory chain. Its conservation across bacterial species such as *E. coli*, *S. flexneri* (*Shigella flexneri*), and *Y. pestis* (*Yersinia pestis*) implies its critical role in bacterial physiology [55,56]. Although terminal oxidase activity was unaffected by the *nuoN* mutation (Appendix A), our results clearly showed that NAD^+^ was increased in the mutant of *PA2649* (*nuoN*). The absolute levels of NAD^+^ in cells are important since they directly impact an organism’s capacity to generate energy and maintain metabolic homeostasis. Recent work has reported that the expression of virulence factors in *P. aeruginosa* is regulated by central metabolism [19]. Consistent with this, the ratio of NAD^+^/NADH in ∆*PA2649* (∆*nuoN*) was increased relative to the wild type and revertant strain (Figure 9A,B). Those results indicated that the mutation evidently influenced bacterial metabolism, which likely accounts for the observed gene expression changes. While a coherent picture is emerging, many fundamental questions about the regulatory link between NADH dehydrogenases and virulence activation remain unresolved.

In summary, our findings demonstrate that the *PA2649* (*nuoN*) mutation induces *exoS* overexpression and cytotoxicity through the Gac-Rsm-Exs pathway. Additional virulence-related phenotypes in the *nuoN* mutant, including motility, adhesion, pyocyanin production, and antibiotic resistance, are modulated independently of the Gac-Rsm-Exs axis, suggesting that NADH dehydrogenase influences virulence through complex metabolic adjustments. These findings underscore the critical role of *PA2649* in coordinating virulence regulation and antibiotic resistance in *P. aeruginosa*, offering insights that may inform the development of novel therapeutic strategies targeting this multifunctional pathway. Further research is needed to clarify how central metabolic changes associated with NADH dehydrogenase activity contribute to virulence regulation.

## Figures and Tables

**Figure 1 microorganisms-13-00392-f001:**
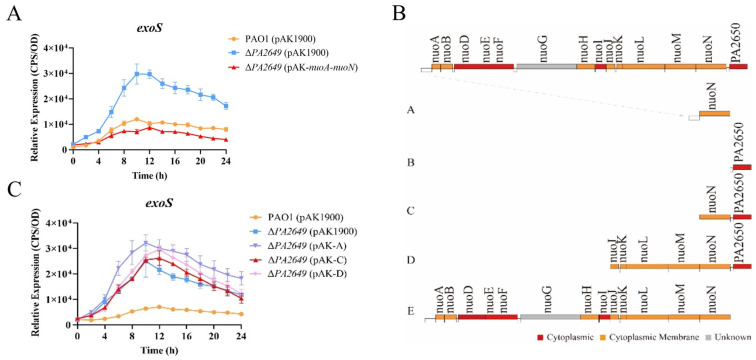
Effect of the *PA2649* mutation on *exoS* expression. (**A**) Expression levels of *exoS* were measured using a chromosomally integrated CTX-*exoS* reporter fusion under T3SS-inducing conditions in wild-type PAO1, the *PA2649* mutant, and the complementary strain. (**B**) Expression levels with additional revertant plasmids carrying different genes. (**C**) Expression of *exoS* in the *PA2649* mutant with various revertant plasmids. Fragment A: This construct includes the *nuoN* coding sequence along with its predicted native promoter of the whole operon. Fragment B: This construct contains the PA2650 coding sequence along with its native promoter. Fragment C: This construct consists of both *nuoN* and PA2650. Fragment D: This fragment spans a larger region, including *nuoJ* through PA2650. Fragment E: This construct contains the predicted full operon with a native promoter, covering *nuoA* to *nuoN*. Results represent the averages of triplicate experiments, with error bars indicating standard deviations. The assay was independently repeated three times. Cps: counts per second.

**Figure 2 microorganisms-13-00392-f002:**
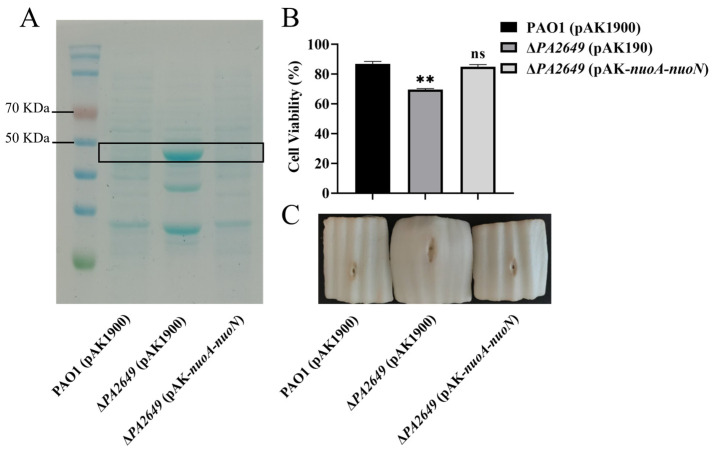
Characterization of T3SS effectors and cytotoxicity assays. (**A**) Secreted T3SS effectors in T3SS-inducing medium were analyzed by SDS-PAGE. Culture supernatants of various strains after 6 h of growth in T3SS inducing medium were precipitated by TCA and analyzed by SDS-PAGE, followed by staining with Coomassie blue. Bands corresponding to ExoS effector are indicated. The black box marks the area corresponding to the proposed proteins on the gel. (**B**) Cytotoxicity assays were performed on A549 cells infected with the supernatant and quantified by the CCK-8 assay. Overnight cultures were centrifuged, and bacterial supernatants were filtered. Equal volumes of supernatants were incubated with A549 cells for 3 h, followed by cytotoxicity analysis. (**C**) Evaluation of necrosis caused by PAO1, *PA2649* mutants, and the complementary strain in a Chinese cabbage model. Results represent the averages of triplicate experiments, with error bars indicating standard deviations. Statistical significance was assessed using a two-tailed unpaired *t*-test. Statistical significance is indicated as ns (not significant), *p* > 0.05, ** *p* < 0.01.

**Figure 3 microorganisms-13-00392-f003:**
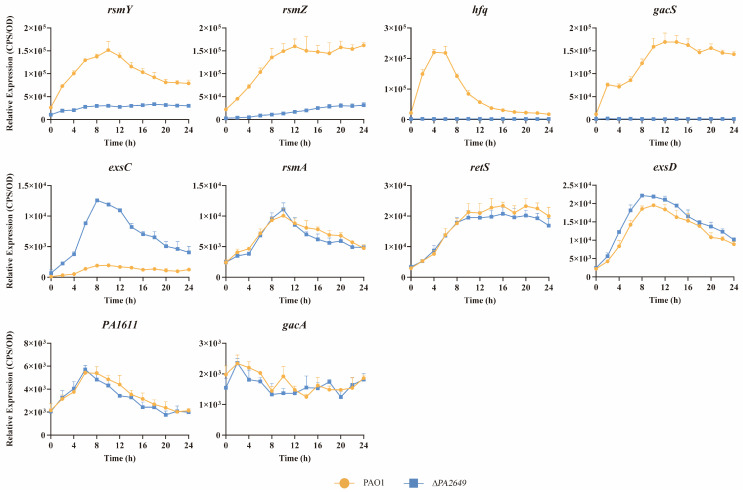
Impact of *PA2649* mutation on expression of T3SS-regulated genes in *P. aeruginosa*. The expression levels of T3SS-regulated genes (*rsmY*, *rsmZ*, *hfq*, *gacS*, *exsC*, *rsmA*, *retS, exsD*, *PA1611*, and *gacA*) were analyzed in *PA2649* mutants. Overnight, different bacterial cultures were inoculated at 1% into fresh LB medium containing appropriate antibiotics and incubated for 3 h. For measurement, 95 µL of LB and 5 µL of bacterial culture were added to each well of a black 96-well plate with a clear bottom, mixed thoroughly, and topped with 50 µL of paraffin oil to prevent evaporation. Luminescence (cps) and bacterial growth (OD_600_) were recorded every 2 h over a 24 h period using a multimode microplate reader, as previously described. Results are presented as the averages of triplicate experiments, with error bars indicating standard deviations.

**Figure 4 microorganisms-13-00392-f004:**
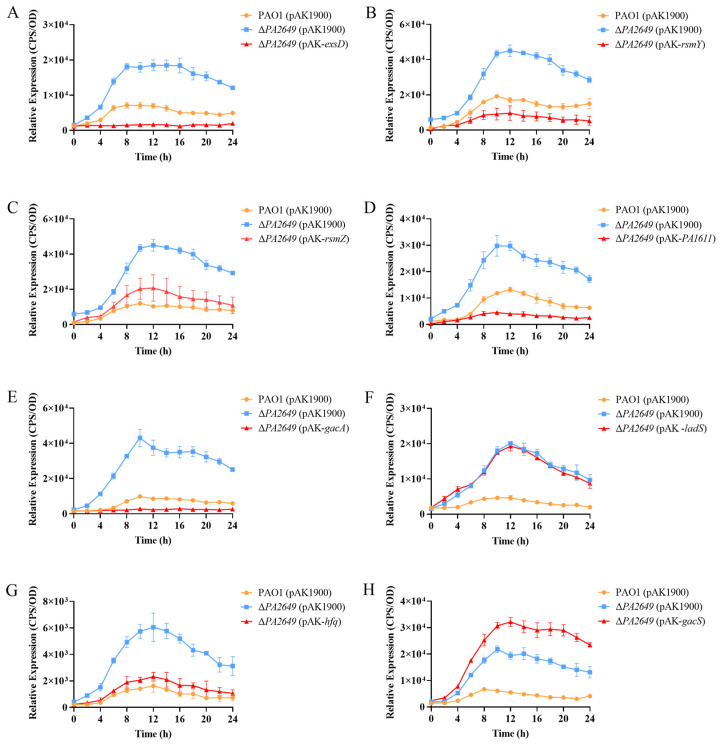
Impact of overexpression of T3SS-regulatory genes on *exoS* expression in *PA2649* mutants. The expression of *exoS* was measured using a chromosomally integrated CTX-*exoS* reporter fusion. Overexpression of T3SS-regulatory genes, which are known to influence the type III secretion system, was analyzed: (**A**) *exsD*; (**B**) *rsmY*; (**C**) *rsmZ*; (**D**) *PA1611*; (**E**) *gacA*; (**F**) *ladS*; (**G**) *hfq*; (**H**) *gacS*. Overnight, different bacterial cultures were inoculated at 1% into fresh LB medium containing appropriate antibiotics and incubated for 3 h. For measurement, 95 µL of LB and 5 µL of bacterial culture were added to each well of a black 96-well plate with a clear bottom, mixed thoroughly, and topped with 50 µL of paraffin oil to prevent evaporation. Luminescence (cps) and bacterial growth (OD_600_) were recorded every 2 h over a 24 h period using a multimode microplate reader, as previously described. Results are presented as averages of triplicate experiments, with error bars indicating standard deviations.

**Figure 5 microorganisms-13-00392-f005:**
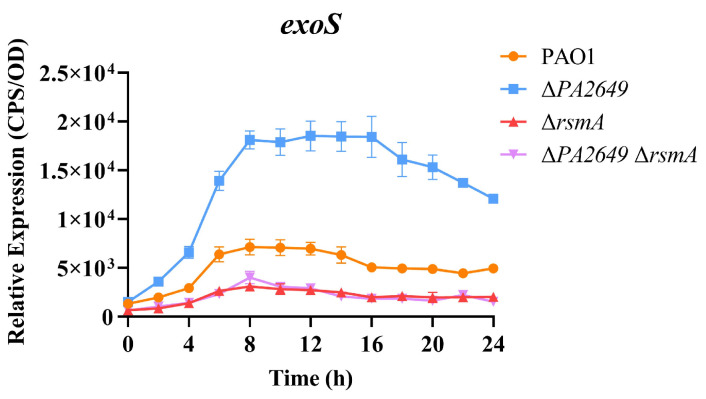
Impact of *rsmA* deletion in the *PA2649* mutant background on *exoS* expression. The expression levels of *exoS* were analyzed in ∆*rsmA*∆*PA2649* double mutants. Overnight, different bacterial cultures were inoculated at 1% into fresh LB medium containing appropriate antibiotics and incubated for 3 h. For measurement, 95 µL of LB and 5 µL of bacterial culture were added to each well of a black 96-well plate with a clear bottom, mixed thoroughly, and topped with 50 µL of paraffin oil to prevent evaporation. Luminescence (cps) and bacterial growth (OD_600_) were recorded every 2 h over a 24 h period using a multimode microplate reader, as previously described. Results are presented as the averages of triplicate experiments, with error bars indicating standard deviations.

**Figure 6 microorganisms-13-00392-f006:**
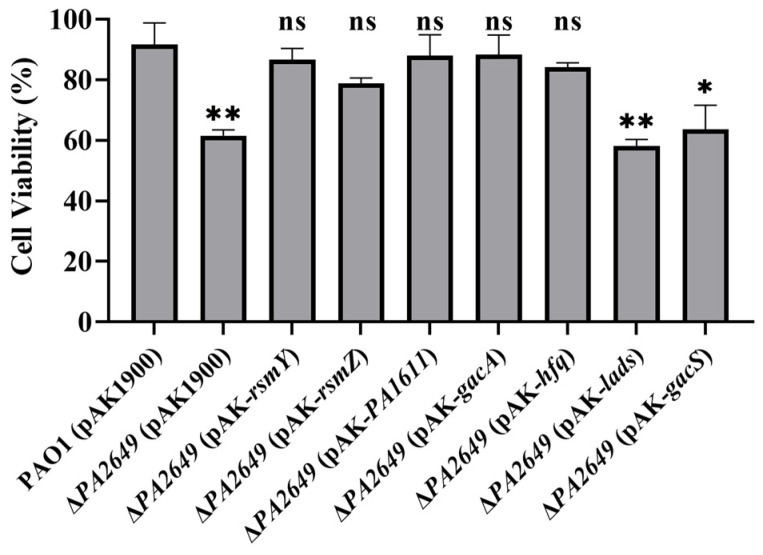
Cytotoxicity assay of *PA2649* mutants overexpressing T3SS-regulatory genes in A549 cells. A549 cells were infected with the supernatants from *PA2649* mutants overexpressing *PA1611*, *hfq*, *gacA*, *rsmY*, *rsmZ*, *ladS*, and *gacS* for 3 h. Bacterial cytotoxicity was assessed using the CCK-8 assay. Overnight, different cultures were centrifuged, and bacterial supernatants were filtered. Equal volumes of supernatants were incubated with A549 cells for 3 h, followed by cytotoxicity analysis. Data represent the results from three independent experiments. Statistical significance was assessed using a two-tailed unpaired *t*-test. Statistical significance is indicated as ns (not significant, *p* > 0.05), * *p* < 0.05; ** *p* < 0.01.

**Figure 7 microorganisms-13-00392-f007:**
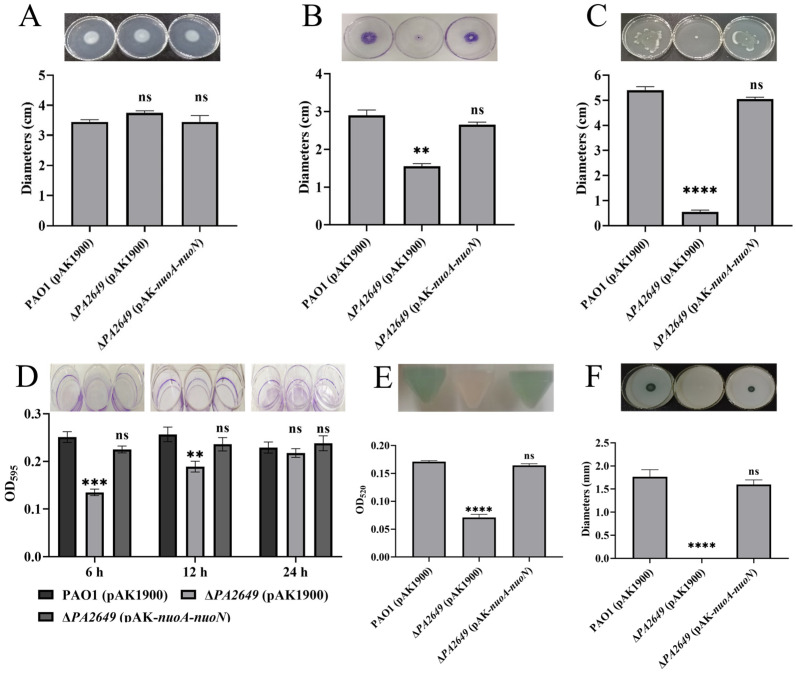
Effects of *PA2649* mutants on the production of virulence factors in *P. aeruginosa*. The influence of *PA2649* mutants on various virulence factors was assessed: (**A**) swimming motility; (**B**) twitching motility; (**C**) swarming motility; (**D**) biofilm formation; (**E**) pyocyanin production; (**F**) protease activity. Statistical significance was assessed using a two-tailed unpaired *t*-test. Statistical significance is indicated as ns (not significant, *p* > 0.05), ** *p* < 0.01, *** *p* < 0.001, and **** *p* < 0.0001.

**Figure 8 microorganisms-13-00392-f008:**
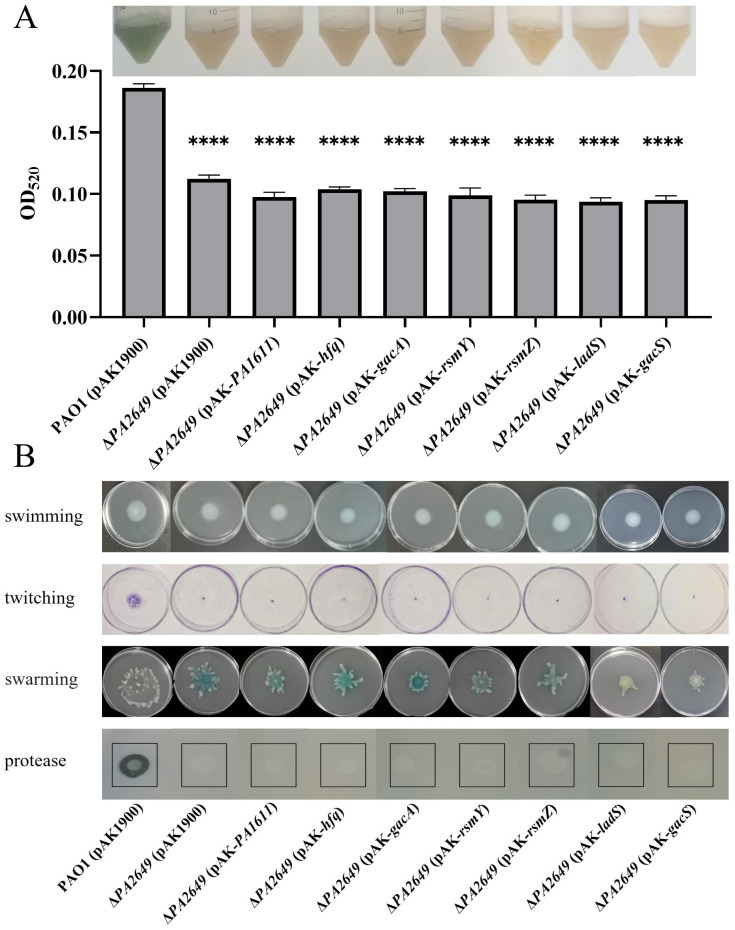
Impact of overexpression of T3SS-regulatory genes on virulence factor production in *PA2649* mutants of *P. aeruginosa*. The effects of overexpressing *PA1611*, *hfq*, *gacA*, *rsmY*, *rsmZ*, *ladS*, and *gacS* in *PA2649* mutants on the production of virulence factors were assessed: (**A**) pyocyanin; (**B**) swimming, twitching, swarming, and protease activity. The black box in the figure highlights the specific area of interest. Statistical significance was assessed using a two-tailed unpaired *t*-test. Statistical significance is indicated as **** *p* < 0.0001.

**Figure 9 microorganisms-13-00392-f009:**
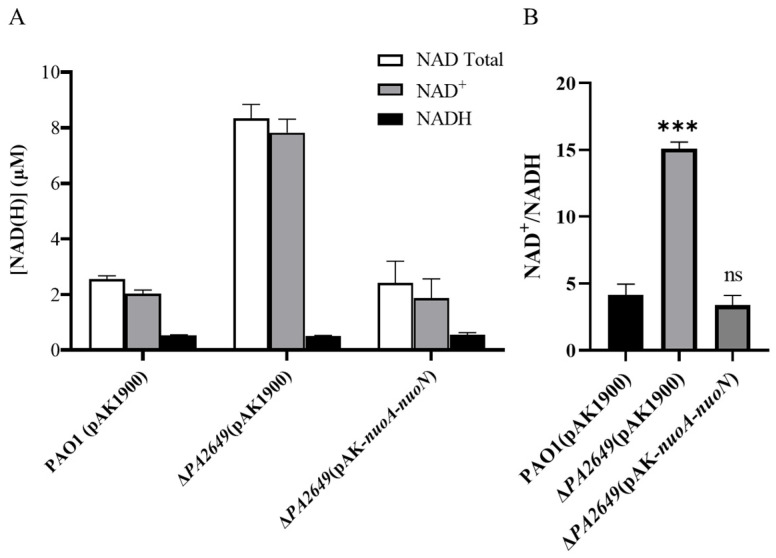
Levels and ratios of NAD⁺ and NADH in *P. aeruginosa*. (**A**) Total concentrations of NADH and NAD⁺ in different strains cultured under the conditions described in the methods section. (**B**) Corresponding NADH/NAD⁺ ratios for the same strains under identical culture conditions. Bacterial strains were cultured overnight in fresh LB medium for approximately 12 h. The OD_600_ of the bacterial cultures was adjusted to the same level before processing. NADH and NAD^+^ were extracted using the Beyotime Enhanced NAD^+^/NADH Assay Kit (WST-8), following the manufacturer’s instructions. Statistical significance was assessed using a two-tailed unpaired *t*-test. Statistical significance is indicated as ns (not significant, *p* > 0.05), *** *p* < 0.001.

**Figure 10 microorganisms-13-00392-f010:**
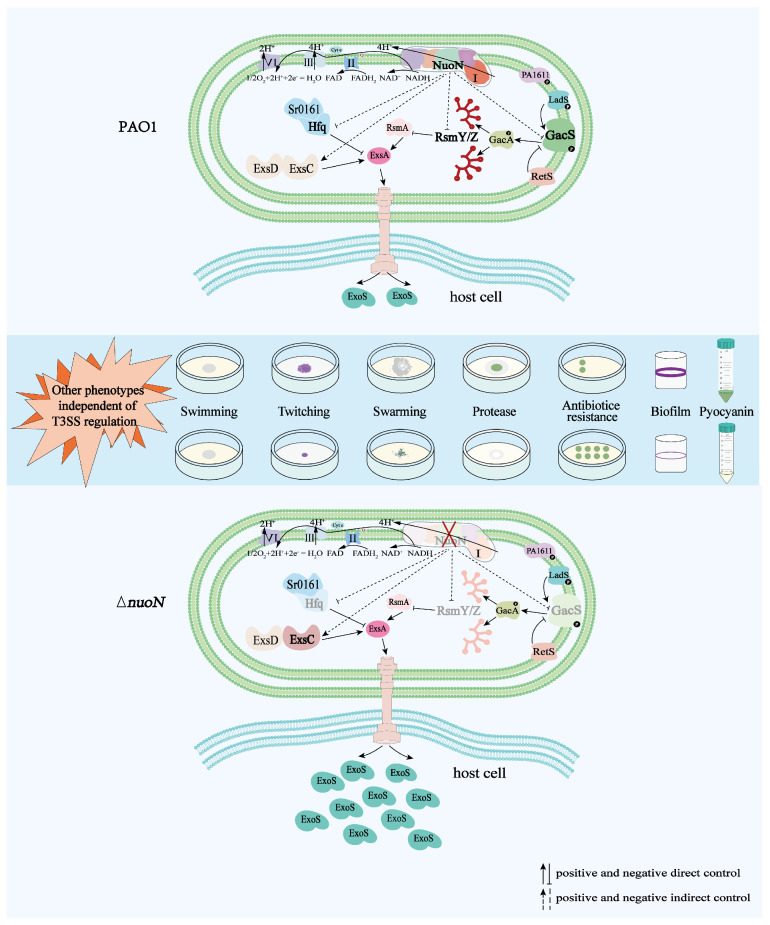
Proposed model illustrating the regulation of bacterial virulence traits by *PA2649* (*nuoN*) in *P. aeruginosa*. Compared to PAO1, the expression of *gacS*, *rsmYZ*, and *hfq* is downregulated, while the expression of *exsC* is upregulated in the ∆*nuoN* mutant, leading to increased secretion of ExoS. Additionally, phenotypic changes such as altered pyocyanin production, initial biofilm formation, motility, antibiotic resistance, and protease production were observed in the ∆*nuoN* mutant. The mutant strain exhibits other phenotypes that are regulated by pathways independent of the T3SS regulatory system.

**Table 1 microorganisms-13-00392-t001:** Minimal Inhibitory Concentrations (MICs) of Different Antibiotics for Various *Pseudomonas aeruginosa* Strains.

Strains	Antibiotics (μg/mL)					
	Gen	Kan	Chl	Strep	Cip	PMB
PAO1 (pAK1900)	1.5	2500	120	15	0.3125	3
Δ*PA2649* (pAK1900)	3	2500	480	15	0.625	3
Δ*PA2649* (pAK-*nuoA*-*nuoN*)	1.5	2500	120	15	0.3125	3
Δ*PA2649* (pAK-*PA1611*)	3	\	480	\	0.625	3
Δ*PA2649* (pAK-*rsmY*)	3	\	480	\	0.625	3
Δ*PA2649* (pAK-*rsmZ*)	3	\	480	\	0.625	3
Δ*PA2649* (pAK-*hfq*)	3	\	480	\	0.625	3
Δ*PA2649* (pAK-*gacA*)	3	\	480	\	0.625	3
Δ*PA2649* (pAK-*gacS*)	3	\	480	\	0.625	3
Δ*PA2649* (pAK-*ladS*)	3	\	480	\	0.625	3

## Data Availability

The original contributions presented in this study are included in the article/Appendix A. Further inquiries can be directed to the corresponding author(s).

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
