# Peer review of "Uncovering the Multifaceted Role of PA2649 (nuoN) in Type III Secretion System and Other Virulence Production in Pseudomonas aeruginosa PAO1"

_microorganisms, 2025, doi:10.3390/microorganisms13020392_

Round 1
Reviewer 1 Report
Comments and Suggestions for Authors
In this work, Chen et al. partially characterize a mutant of the PA2649 gene (nouN), which encodes for a NADH dehydrogenase related to the delivery of electrons to the respiratory chain. The regulatory effect of nouN on the expression of the exoS toxin by the type III secretion system is reported. The regulation is reported to be mediated by the Gac-Rsm regulatory pathway. Collaterally, the effect on other phenotypes, such as mobility, biofilm formation, synthesis of proteases and pyocyanins, and antibiotic resistance, is described preliminarily.
In general, is an acceptable work. Some observations are listed below; once the comments are addressed, the new version could be accepted for publication.
Major concerns
According to the site-directed mutagenesis strategy (using the pEX vector), the generated mutation should be non-polar. However, complementation analyses show a polar effect of the mutation. How can this be explained? It would be undesirable to perform experiments that could determine the impact of the polarity of the mutation. Would a non-polar mutant have the same phenotype?
Given that these proteins are a regulatory pair, how can we explain the different effects of overexpression of gacS and gacA? It is not enough to refer to what has been seen in another model. This has to be discussed, as it is already known that this two-component system is controlled at different levels, including gene expression, interaction with other proteins, dual kinase/phosphatase activity, stoichiometric ratio, etc. Similarly, it is not clear why other elements of the GacS multikinase system (LadS, PA1611) are present.
Were the pools of oxidized, reduced cofactors measured in the mutant? Other mutants with similar effects on reductor power would have similar effects on the altered phenotypes in the mutant. It would be desirable to address this situation.
How can we explain why resistance to gentamicin and not to kanamycin is affected if the two antibiotics belong to the same family and share mechanisms of action?
Minor concerns
Genetic complementation involves restoring a phenotype caused by a gene mutation when a wild-type allele of the mutated gene is introduced into the mutant. If other genes are introduced and the phenotype is partially or totally restored, this suggests a functional relationship between the mutated gene and the one being introduced, but this is not genetic complementation but a reversal of the phenotype. It is recommended that the word complementation be corrected throughout the text.
Review using italics to refer to genes and organisms in the text.
The first paragraph (lines 414-432) of the discussion is dispensable; it only describes the results.
Figure 9 is unnecessary; Table 1 is sufficient.
Reviewer 2 Report
Comments and Suggestions for Authors
Titel: Uncovering the Multifaceted Role of P12649 (nuoN) in Type III Secretion System and Other Virulence Production in Paeruginosa aeruginosa PA01
The work of Chen and colleagues investigates an interesting interplay affected by PA2649 in Pseudomonas aeruginosa. They reveal that deletion of PA2649 affects the expression of exoS. They performed additional analyses indicating a relation between PA2649 and range of virulence determinants in this pathogen; however, the following points need to be fully addressed in their study:
A) Major points of concern:
1. The main weak point in this manuscript is being done in only one background of Pseudomonas aeruginosa. Are these findings applicable to other backgrounds? How did the authors exclude any strain-specific phenotypes which could be not the case in other strains?
2. How could the authors explain the enhanced cytotoxicity upon deleting nuoN, despite the reduced expression of virulence determinants/traits?
3. I do not understand whether the authors investigated the release of EXOS in the bacterial supernatants or they measured the activity of exoS in the bacterial cells themselves, because some experiments were done with bacterial cells and others only with supernatants.
B) Minor points of concern:
1. I have no idea what the authors mean by “Paeruginosa aeruginosa PA01” in the title!
2. In line 41: please enumerate some examples to the mentioned chemical signals.
3. In line 75: please indicate which “other global regulatory systems” meant here.
4. In line 84: please modify the sentence “These studies provide a insights into the involvement of NADH related genes in the regulation of virulence”. The sentence is not correct grammatically.
5. General comment 1: please indicate the source or the reference of every used chemical or media in this study e.g. LB medium or PIA or the plates used for measuring the gene expression.
6. In line 156: how long time was needed to reach an OD600 of 1.0. Please indicate in the main text.
7. How did the authors ascertain that the retarded virulence traits were real and not attributed to a defective growth phenotype in the tested strains?
8. In the Chinese Cabbage Infection Assay: please indicate how the infection/damage degree was estimated? Did the authors include control experiments here?
9. In the Swimming, Swarming and Twitching Motility Assay: please indicate how the diameter of zones was estimated. How did the authors use crystal violet?
10. In figure 1: I am confused with the term fragments A-E. Also, the figure itself is not helpful. Please provide a detailed description of these fragments in the main text.
11. General comment 2: in all figures, please add a brief description of the experiment itself. It would be helpful too to mention which statistical used was used to analyse each figure.
12. For the complementation strategies used. The authors need to explain whether they used in-cis order trans complementation strategy.
13. In line 284-285: describing the results at this point so is misleading. It would be enough at this point to mention that this phenotype is attributed to the deletion of nuoN.
14. Despite that the CTX-based reporter fusion experiments are very interesting; however, the qRT-PCR would be much more reliable than measuring the luminesce in a plate reader. At least the expression of exoS in the PA2649-mutant should be measured by qRT-PCR. Please also indicate how the CPS/OD values were calculated.
15. In line 317: the assumption that the expression of GacA was reduced in the mutant strain is incorrect and does not fit with the results in figure 3.
16. In figure 7: the data of the biofilm formation do not match with the data in figure 1. The expression vales of exoS at 6 and 24 hours for the WT, PA2649-mutant and complemented strain looks identical. Only at the time frame of 6-12 hours the expression values at the mutant strain are much higher than the two other strains. Why the biofilm formation at 24 hours showed no difference between the three strains; despite the differences were clear at 6 hours? Could be this phenotype exoS-independent?
17. In figure 7: it is difficult to imagine that the deletion of PA2649 completely aborted the proteolytic activity as indicated in figure 7F. This also does not match with figure 2A where several other smaller proteins are detected in the PA2649-mutant strain.
18. In the discussion part: it would be worth to include some information related to the importance of nuoN as an effector of resistance of Pseudomonas aeuroginosa to penicillin-based therapies. This would provide somehow completed picture about the antibiotic resistance profile affected by nuoN.
19. Why did the authors choose to isolate the EXOS proteins after 6 hours, despite the data in figure 1 showed that the higher expression values were detected after 9 and 12 hours?

Round 2
Reviewer 1 Report
Comments and Suggestions for Authors
I consider that the observations and suggested corrections were attended to favorably; therefore, this version of the manuscript can be published.